# Investigating the Geographic Disparities of Amenable Mortality and Related Ambulance Services in Hungary

**DOI:** 10.3390/ijerph18031065

**Published:** 2021-01-25

**Authors:** Máté Sándor Deák, Gábor Csató, György Pápai, Viktor Dombrádi, Attila Nagy, Csilla Nagy, Attila Juhász, Klára Bíró

**Affiliations:** 1Faculty of Public Health, University of Debrecen, 4028 Debrecen, Hungary; deak.mate@sph.unideb.hu (M.S.D.); nagy.attila@sph.unideb.hu (A.N.); 2Doctoral School of Health Sciences, University of Debrecen, 4028 Debrecen, Hungary; 3Hungarian National Ambulance Service, 1055 Budapest, Hungary; csato.gabor@mentok.hu (G.C.); papai.gyorgy@mentok.hu (G.P.); 4Health Services Management Training Centre, Faculty of Health and Public Administration, Semmelweis University, 1125 Budapest, Hungary; dombradi.viktor@emk.semmelweis.hu; 5Department of Public Health, Government Office of the Capital City Budapest, 1138 Budapest, Hungary; nagy.csilla@kmr.antsz.hu (C.N.); juhasz.attila@kmr.antsz.hu (A.J.); 6Department of Health Systems Management and Quality Management for Health Care, Faculty of Public Health, University of Debrecen, 4032 Debrecen, Hungary

**Keywords:** amenable mortality, acute myocardial infarction, hemorrhagic stroke, ischemic stroke, ambulance service, Hungary

## Abstract

The aim of this study was to investigate how amenable mortality and related ambulance services differ on a county level in Hungary. The differences in mortality rates and ambulance services could identify counties where stronger ambulance services are needed. The datasets for 2018 consisted of county level aggregated data of citizens between the ages 15–64. The study examined how both the mortality rates and the ambulance rescue deliveries differ from the national average. The analyses were narrowed down to disease groups, such as acute myocardial infarction, hemorrhagic and ischemic stroke. Inequalities were identified regarding the distribution of number of ambulance deliveries, several counties had rates more than double that of the national average. For both mortality and ambulance services some of the counties had significantly better results and others had significantly worse compared to the national average. In Borsod-Abaúj-Zemplén county’s case, hemorrhagic stroke mortality was significantly higher (1.73 [1.35–2.11]), while ambulance deliveries were significantly lower (0.58 [0.40–0.76]) compared to the national average. The research has shown that regarding the investigated mortality rates and ambulance services there are considerable differences between the counties in Hungary. In this regard policy makers should implement policies to tackle these discrepancies.

## 1. Introduction

Amenable mortality is a lesser-known factor even though it could provide an accurate picture about health care systems, as it could give additional information to health care providers and decision makers at multiple levels [1,2]. “*A death can be considered as amenable if it could have been avoided through optimal quality health care*” [3]. It should not be confused with preventable mortality, which is *”…broader and includes deaths which could have been avoided by public health interventions focusing on wider determinants of public health, such as behavior and lifestyle factors, socioeconomic status and environmental factors*” [3]. The combination of amenable mortality and preventable mortality is called avoidable mortality.

In 2018 a set of indicators were created by a Spanish research team to measure amenable mortality [4]. In these indicators they only included disorders which can be diagnosed by the symptoms alone, as well as if patients would get adequate treatment in time, this could considerably improve their health.

Regarding amenable mortality Hungary is one of the worst among EU countries [5]. In November of 2010 a Hungarian research was published about the trends of mortality amenable by health care. The article stated that “*the years of potential life loss of males were 7207 (per 100,000) and respectively for females this value was 3870 (per 100,000) in Hungary in 2006. The amenable mortality is a significant contributor to years of potential life lost despite its decreasing trend. The amenable mortality accounted for approximately one-third of the males’ and the females’ years of potential life lost*” [6]. Thus, it can be concluded that if we could decrease these mortality numbers, it would be a great aid in reducing avoidable deaths [7,8,9,10]. Also, by the latest accessible Eurostat data, Hungary is among the most afflicted EU countries in cardiovascular diseases. After Lithuania, Hungary has some of the highest standardized death rates for ischemic heart diseases. Hungary also has the largest number of patients with self-reported hypertensive diseases in the whole EU [11].

Furthermore, the results of this aforementioned study show that about a quarter of premature mortality in Hungary is because of amenable deaths due to lack of proper health care [6]. Therefore, it is essential to find out the causes and formulate measures for decision-makers to reduce the number of avoidable mortalities as much as possible.

The care of pre-hospital acute cases is the primary task of the ambulance service. The survival and recovery of patients depends to a large extent on when they get help [12,13]. In many cases even laic assistance can considerably increase survival, starting out with simple bleeding suppression, plain coverings, through the fixation of body parts up to resuscitation. However, the result depends mainly on the time of initiation of the advanced level professional assistance. In most cases, this assistance depends on the competent ambulance service, which is why it is important to examine the performance of the particular service providers. At the international level, several researches were done on this topic, but without exception, organizations operating on a market or semi-market basis have been studied mainly in Anglo-Saxon countries [14,15,16]. Limited number of researches were conducted on ambulance services in Hungary and all of them have focused on a specific area of health care and have not addressed the issue of the health care organization’s own activities [13].

In Hungary the National Ambulance Service (Országos Mentőszolgálat—OMSz) has been operating for more than 70 years [17]. The organization is completely government funded and all the ambulance deliveries are conducted by it. Hungary has 16 PCI (percutaneous coronary intervention) centers, 39 stroke centers, and out of all the centers six also have mechanical thrombectomy capability. In the past few decades the National Ambulance Service has been informing and educating the citizens as well as the medical providers in Hungary to call immediately an ambulance if an acute coronary syndrome or stroke is suspected in order to minimize the pre-hospital delay of the reperfusion treatment. By using regularly updated standard operational procedures (SOP) all over the country the National Ambulance Service tries to provide the same quality of service, resulting in equal recovery chances to all the patients.

The examination of the performance of the Hungarian ambulance service can provide clear guidelines for health policy decision-makers, as it provides in an international comparison a unique and complete coverage of ambulance activity with a single national level provider, which is entirely publicly funded, and has uniform approaches and procedures [18].

As for amenable mortality, the source of error may not only be in pre-hospital care but also in institutional care and acute care. Nevertheless, the performance of the ambulance service can be essential in terms of both time factor and quality of care [19]. Beyond these factors, however, the service provider must strive to reduce geographic inequality as much as possible. This can only be achieved, if the rate of ambulance deliveries—which is a reflection of the availability of services provided by the National Ambulance Service—and the health needs of the general population are in synch with each other.

The goal of the research was to examine the performance of an ambulance service in circumstances where it is responsible for rescuing more than 9 million residents and ameliorating outcome indicators can achieve considerable improvements in avoidable early deaths. In this study, in the light of amenable mortality, we aimed to identify possible geographic inequalities regarding ambulance rescue in Hungary. Such results could help decision-makers identify those areas within the country, where intervention is needed to tackle inequalities in care.

## 2. Materials and Methods

For this study an ethical approval was obtained from the Scientific Research and Ethics Committee of the Medical Research Council in Hungary. Ethical authorization number: ETT-TUKEB 41880-2/2019/EKU. In our research, we examined the number of deaths among the Hungarian population, and the ambulance rescue data. The former data was obtained from the Hungarian Central Statistical Office (Központi Statisztikai Hivatal—KSH) and the latter from the Hungarian National Ambulance Service. All the data obtained included aggregated data of residents between the age 15–64 in the year 2018. Residents over the age of 64 were excluded in our study as we focused on amenable mortality among adults. The concept of avoidable mortality has undergone significant changes in content and terminology since its inception in 1976, including a change in the upper limit of the initial limit from 64 to 74 years [10,20]. The use of a higher upper age limit was justified by the steady increase in life expectancy at birth and also by the fact that better results are being achieved at older age using innovative health interventions [10,20]. Evidence shows however, that there are significant differences in the life expectancies of people in different European countries. Thus, for example, compared to the life expectancy of the population of the European Region, the Hungarian population lagged by almost 3 years in 2015 [21]. Moreover, in Hungary the gap between the life expectancy of men and women is also considerable. The difference was about 8.8 years in 2015 [22], but despite improving trends, it is still 6.47 years in 2019 (men: 72.86 years; women: 79.33 years) [23]. In addition, the outstandingly poor health outcomes of the Hungarian population, especially in the “early death stage of life (in the 25–64 age group)”, is also well known [21,24,25]. Consequently, due to the considerable lag in the life expectancy of the Hungarian population and its high early mortality, in the present analysis we decided to examine the deaths related to health services that occurred in the 15–64 age group.

Furthermore, the minimum age limit was lowered from 18 to 15 years so that the existing age distributions could be grouped into equal five years. We have narrowed our data to acute myocardial infarction (AMI) (ICD10: I21, I22), hemorrhagic stroke (ICD10: I60, I61, I62) and ischemic stroke (ICD10: I63, I64) because among the amenable mortalities, these are the diseases that can be examined directly in the ambulance service with the least influencing factors. In all three cases, the ambulance service is prepared to apply appropriate intervention on site, and time factor is particularly important for these diseases. National data on ambulance rescue numbers and mortality were received on a county level (Nomenclature of Territorial Units for Statistics at level 3—NUTS 3).

The two original databases obtained included raw indicators without any filtering. Such indicators cannot be compared as different distributions of age and gender would lead to misleading results in the study. Therefore, indirect standardization was performed, where the reference base was the national ambulance rescue frequency and the national mortality. As a result, we managed to obtain the mortality rate of each county, which allowed the mapping of territorial inequalities across the country.

To calculate the disparities between the counties standardized mortality of the given disease and standardized ambulance rescue for the given cases were used. The mortalities are standardized mortality ratio (SMR) with 95% confidence interval. Microsoft Excel was used for all calculations.

## 3. Results

In 2018 6,729,449 Hungarians were between the ages 15 and 65. Overall 29,154 people died between the same age group and 369,672 deliveries were carried out by the Hungarian National Ambulance Service (Table 1).

Table 2 depicts AMI-related ambulance deliveries relative to AMI mortality. The results show significant differences from the country average in case of both AMI mortality and AMI related ambulance deliveries were found in four counties. Significantly lower AMI mortality has been found in Baranya (0.55 [0.36–0.74]) and Hajdú-Bihar (0.64 [0.47–0.74]). Furthermore, in these two counties the AMI related ambulance deliveries were significantly lower too. We found significantly higher AMI mortality in Borsod-Abaúj-Zemplén, (1.28 [1.06–1.51]) and Jász-Nagykun-Szolnok (1.32 [1.02–1.61]). However, in these counties the AMI related ambulance deliveries were significantly higher than the national average.

The results of hemorrhagic stroke-related ambulance deliveries relative to hemorrhagic stroke mortality show significant differences from the country average in case of both hemorrhagic stroke mortality and hemorrhagic stroke-related ambulance deliveries were found in five counties. Significantly lower hemorrhagic stroke mortality has been found in the capital, Budapest (0.67 [0.52–0.82]), in Pest county (0.76 [0.58–0.95]) and Vas (0.49 [0.17–0.81]). In these counties the hemorrhagic stroke-related ambulance deliveries were significantly lower also. We found significantly higher hemorrhagic stroke mortality in Szabolcs-Szatmár-Bereg (1.41 [1.04–1.78]) with a significantly higher hemorrhagic stroke-related ambulance deliveries as well. In the case of Borsod-Abaúj-Zemplén county we found significantly higher hemorrhagic stroke mortality, (1.73 [1.35–2.11]), but the hemorrhagic stroke-related ambulance deliveries were significantly lower (0.58 [0.40–0.76]) (Table 3).

Table 4 shows the relative values of ischemic stroke mortality and ischemic stroke- related ambulance deliveries. The results show significant differences from the country average in case of both ischemic stroke mortality and ischemic stroke-related ambulance deliveries in eleven counties. Significantly lower ischemic stroke mortality has been found in Budapest (0.84 [0.81–0.86]), in Pest (0.93 [0.90–0.97]) and Zala county (0.90 [0.84–0.96]). In these counties the ischemic stroke-related ambulance deliveries were also significantly lower. Significantly higher ischemic stroke mortality was identified in Békés (1.11 [1.04–1.11]), Borsod-Abaúj-Zemplén (1.26 [1.21–1.31]), Heves (1.13 [1.06–1.20]), Jász-Nagykun-Szolnok (1.21 [1.15–1.28]), Nógrád (1.21 [1.12–1.29]) and Szabolcs-Szatmár-Bereg (1.10 [1.05–1.15]). In these counties the ischemic stroke-related ambulance deliveries were significantly higher too, relative to the national average. In the cases of Győr-Moson-Sopron and Hajdú-Bihar counties the ischemic stroke mortality was significantly lower (0.90 [0.85–0.95] and 0.89 [0.85–0.94]), but the ischemic stroke-related ambulance deliveries were significantly higher as well.

## 4. Discussion

This is the first study to examine the relationship between amenable mortality and ambulance deliveries at the county level in Hungary. The results show that there are considerable differences between the counties, as many of them had mortality and delivery rates near double that of the national average, although the differences vary depending on the type of disease. There are geographic inequalities regarding mortality in Hungary [26], thus, similar results for the ambulance care were to be expected. The most noteworthy difference was found in the case of Borsod-Abaúj-Zemplén county, where significantly higher hemorrhagic stroke mortality was found, but the hemorrhagic stroke-related ambulance deliveries were significantly lower. However, Borsod-Abaúj-Zemplén is one of the most socially and economically deprived counties in Hungary [27], and this might have affected the results as well. Nevertheless, this information can be important in itself for the decision makers, as it indicates that the allocation of resources in regards of ambulance services can be further optimized. The importance of amenable mortality must come to the forefront of decision-makers, as there are opportunities to avoid such deaths [7,8,9]. In this endeavor ambulance deliveries should be considered as well as identified service inequalities should be tackled.

This study also draws the attention of the management of the National Ambulance Service to further emphasize the utilization of check-list type disease debriefing and the supervision of this procedure during the training of rescue managers and those taking part in rescue operations in order to be able to make appropriate decisions on the place of dispatch and the level of competency needed from the dispatched rescue unit, thus, ensuring the highest quality of care possible.

Furthermore, based on the results of the analysis the operative team of the National Ambulance Service can formulate a proposal on the appropriate allocation of ambulance units to the top management, in order to aid the implementation of a rational and professional vehicle reallocation.

Within the limits of our study, it should be mentioned that the multifactorial etiology of the cardiovascular diseases studied was not considered in the study. Our mortality and ambulance delivery results at the county level not necessarily mean the same outcomes in individuals. The detected results by our investigation do not prove a causal relationship between mortality and ambulance deliveries. The study was conducted at the county level as the data of the Hungarian ambulance service were available in this form, so the study could not analyze the municipalities according to the degree of urbanization and their geographical location. An additional limitation is that although the Hungarian ambulance service provides uniform coverage of patient transportation in the country, the related healthcare providers in the geographical areas may not be of exactly the same professional standard and equipment throughout the country, which may affect avoidable mortality rates. Thus, the aim of our study was to present a result that draws attention to the importance of the efficient operation of the ambulance service and the existence of differences that can be reduced.

## 5. Conclusions

The principle of equal healthcare access is an important health policy objective [28]. Thus, Hungarian decision-makers should create policies and an overall strategy based on the obtained results. For example, better allocation of resources between the counties, the expansion of priority units in some counties, or creating efficient health screenings in the underperforming areas.

In order to reduce inequalities, an important action would be to train and educate the general population regarding the symptoms of various diseases, the encouragement of making rescue calls and providing first aid. The feedbacks of these actions should be shared with both the ambulance service’s rescue and management staff and, if necessary, professional training should be provided to achieve better results. Based on the results of this study, the National Ambulance Service plans to introduce a new, uniform and professional rescue manager interview protocol, as well as to issue a new organization form regarding the vehicle fleet, location and necessary modification of ambulance vehicles.

Such implemented policies should be examined, and the impact of the interventions should be evaluated as well. Lessons learned from successful interventions can and should be shared with rescue services in other countries. Furthermore, we encourage other researchers to conduct similar studies, as it is probable, that similar differences can be identified.

## Figures and Tables

**Table 1 ijerph-18-01065-t001:** Percentage of the overall Hungarian population, mortality and ambulance deliveries between the ages 15–64 years in 2018 at the county level.

County (NUTS 3)	Population Between 15–64 Years			All Mortality for 15–64 Years Old			All Ambulance Deliveries for 15–64 Years Old		
	Female	Male	Total	Female	Male	Total	Female	Male	Total
Total (*n*)	3,355,836	3,373,614	6,729,449	9770	19,394	29,164	161,414	208,258	369,672
Bács-Kiskun	5.2%	5.3%	5.3%	5.2%	5.8%	5.6%	5.0%	5.1%	5.1%
Baranya	3.9%	3.9%	3.9%	4.4%	4.0%	4.2%	4.9%	4.7%	4.7%
Békés	3.5%	3.6%	3.6%	4.2%	4.1%	4.1%	4.6%	4.5%	4.5%
Borsod-Abaúj-Zemplén	6.8%	6.9%	6.9%	8.4%	8.6%	8.5%	8.7%	7.7%	8.2%
Budapest (capital)	17.2%	16.1%	16.6%	14.8%	12.8%	13.5%	12.8%	13.8%	13.3%
Csongrád	4.1%	4.1%	4.1%	4.5%	3.9%	4.1%	3.3%	3.4%	3.3%
Fejér	4.3%	4.4%	4.3%	4.0%	4.8%	4.5%	4.1%	4.6%	4.4%
Győr-Moson-Sopron	4.5%	4.6%	4.6%	4.0%	4.1%	4.1%	4.3%	5.2%	4.8%
Hajdú-Bihar	5.5%	5.6%	5.5%	4.7%	4.9%	4.8%	7.7%	7.0%	7.3%
Heves	3.0%	3.0%	3.0%	3.6%	3.4%	3.5%	4.2%	3.9%	4.0%
Jász-Nagykun-Szolnok	3.8%	3.9%	3.8%	4.7%	4.7%	4.7%	5.1%	4.5%	4.7%
Komárom-Esztergom	3.1%	3.2%	3.1%	3.5%	3.2%	3.3%	2.7%	2.9%	2.8%
Nógrád	1.9%	2.0%	2.0%	2.1%	2.7%	2.5%	2.1%	2.1%	2.1%
Pest	12.9%	12.9%	12.9%	11.9%	11.1%	11.4%	6.1%	5.8%	6.0%
Somogy	3.2%	3.2%	3.2%	3.4%	3.6%	3.5%	4.0%	3.9%	3.9%
Szabolcs-Szatmár-Bereg	5.9%	6.1%	6.0%	6.0%	6.4%	6.3%	9.9%	9.4%	9.7%
Tolna	2.3%	2.3%	2.3%	2.3%	2.3%	2.3%	2.3%	2.3%	2.3%
Vas	2.5%	2.6%	2.6%	2.2%	3.0%	2.8%	2.7%	3.1%	2.9%
Veszprém	3.5%	3.6%	3.5%	3.4%	3.6%	3.6%	3.3%	3.7%	3.5%
Zala	2.8%	2.8%	2.8%	2.6%	2.9%	2.8%	2.3%	2.4%	2.4%

**Table 2 ijerph-18-01065-t002:** AMI mortality and AMI-related ambulance deliveries in Hungary, 2018.

County (NUTS 3)	AMI Mortality				AMI Related Ambulance Deliveries			
	Case (*n*)	SR *	Lower 95% CI	Upper 95% CI	Case (*n*)	SR *	Lower 95% CI	Upper 95% CI
Bács-Kiskun	67	0.85	0.64	1.05	175	0.79	0.68	0.91
Baranya	33	0.55	0.36	0.74	136	0.82	0.68	0.95
Békés	54	0.97	0.71	1.23	182	1.19	1.02	1.36
Borsod-Abaúj-Zemplén	128	1.28	1.06	1.51	453	1.62	1.47	1.77
Budapest (capital)	236	1.00	0.88	1.13	572	0.86	0.79	0.93
Csongrád	54	0.88	0.64	1.11	126	0.73	0.60	0.86
Fejér	64	0.99	0.74	1.23	327	1.80	1.61	2.00
Győr-Moson-Sopron	55	0.82	0.60	1.04	132	0.70	0.58	0.82
Hajdú-Bihar	51	0.64	0.47	0.82	192	0.86	0.74	0.98
Heves	60	1.32	0.99	1.66	242	1.91	1.67	2.15
Jász-Nagykun-Szolnok	76	1.32	1.02	1.61	195	1.22	1.05	1.39
Komárom-Esztergom	50	1.08	0.78	1.37	120	0.92	0.75	1.08
Nógrád	35	1.14	0.76	1.51	123	1.44	1.19	1.70
Pest	195	1.08	0.93	1.23	181	0.35	0.30	0.40
Somogy	49	0.97	0.70	1.24	106	0.76	0.62	0.91
Szabolcs-Szatmár-Bereg	101	1.20	0.97	1.44	382	1.61	1.45	1.77
Tolna	36	1.00	0.67	1.33	168	1.69	1.44	1.95
Vas	42	1.06	0.74	1.37	63	0.57	0.43	0.71
Veszprém	46	0.83	0.59	1.07	192	1.25	1.07	1.43
Zala	42	0.92	0.64	1.20	71	0.57	0.43	0.70

* SR: Standardized rate.

**Table 3 ijerph-18-01065-t003:** Hemorrhagic stroke mortality and hemorrhagic stroke-related ambulance deliveries in Hungary, 2018.

County (NUTS 3)	Hemorrhagic Stroke Mortality				Hemorrhagic Stroke-Related Ambulance Deliveries			
	Case (*n*)	SR *	Lower 95% CI	Upper 95% CI	Case (*n*)	SR *	Lower 95% CI	Upper 95% CI
Bács-Kiskun	45	1.23	0.87	1.59	48	0.94	0.67	1.20
Baranya	21	0.76	0.43	1.08	172	4.47	3.80	5.14
Békés	34	1.34	0.89	1.79	51	1.45	1.05	1.84
Borsod-Abaúj-Zemplén	80	1.73	1.35	2.11	38	0.58	0.40	0.76
Budapest (capital)	73	0.67	0.52	0.82	82	0.52	0.41	0.63
Csongrád	35	1.23	0.82	1.64	31	0.77	0.50	1.04
Fejér	29	0.97	0.62	1.32	32	0.76	0.50	1.02
Győr-Moson-Sopron	25	0.81	0.49	1.13	45	1.02	0.72	1.32
Hajdú-Bihar	33	0.90	0.59	1.20	41	0.78	0.54	1.02
Heves	24	1.15	0.69	1.61	56	1.91	1.41	2.41
Jász-Nagykun-Szolnok	24	0.91	0.54	1.27	62	1.67	1.25	2.08
Komárom-Esztergom	19	0.88	0.49	1.28	17	0.56	0.29	0.83
Nógrád	13	0.92	0.42	1.42	20	1.02	0.57	1.47
Pest	64	0.76	0.58	0.95	20	0.16	0.09	0.24
Somogy	30	1.29	0.83	1.76	49	1.53	1.10	1.96
Szabolcs-Szatmár-Bereg	55	1.41	1.04	1.78	95	1.69	1.35	2.03
Tolna	11	0.67	0.27	1.06	20	0.88	0.49	1.26
Vas	9	0.49	0.17	0.81	17	0.67	0.35	0.98
Veszprém	31	1.22	0.79	1.65	41	1.16	0.81	1.52
Zala	26	1.25	0.77	1.73	29	1.01	0.65	1.38

* SR: Standardized rate.

**Table 4 ijerph-18-01065-t004:** Ischemic stroke mortality and ischemic stroke-related ambulance deliveries in Hungary, 2018.

County (NUTS 3)	Ischemic Stroke Mortality				Ischemic Stroke-Related Ambulance Deliveries			
	Case (*n*)	SR *	Lower 95% CI	Upper 95% CI	Case (*n*)	SR *	Lower 95% CI	Upper 95% CI
Bács-Kiskun	34	1.04	0.99	1.10	418	0.96	0.94	0.97
Baranya	19	1.03	0.97	1.09	447	1.21	1.19	1.22
Békés	29	1.11	1.04	1.17	364	1.26	1.24	1.28
Borsod-Abaúj-Zemplén	63	1.26	1.21	1.31	719	1.18	1.17	1.19
Budapest (capital)	82	0.84	0.81	0.86	1016	0.82	0.81	0.83
Csongrád	29	0.98	0.93	1.04	157	0.80	0.79	0.82
Fejér	31	1.03	0.97	1.09	397	1.01	0.99	1.02
Győr-Moson-Sopron	21	0.90	0.85	0.95	361	1.05	1.04	1.07
Hajdú-Bihar	21	0.89	0.85	0.94	344	1.31	1.30	1.33
Heves	23	1.13	1.06	1.20	362	1.33	1.31	1.35
Jász-Nagykun-Szolnok	17	1.21	1.15	1.28	366	1.23	1.21	1.25
Komárom-Esztergom	24	1.04	0.98	1.11	353	0.89	0.87	0.91
Nógrád	20	1.21	1.12	1.29	246	1.03	1.01	1.05
Pest	67	0.93	0.90	0.97	578	0.47	0.46	0.48
Somogy	20	1.03	0.97	1.10	426	1.20	1.18	1.22
Szabolcs-Szatmár-Bereg	47	1.10	1.05	1.15	740	1.61	1.59	1.62
Tolna	13	0.93	0.86	1.00	196	0.99	0.97	1.01
Vas	19	1.03	0.96	1.10	232	1.13	1.11	1.15
Veszprém	35	0.95	0.90	1.01	392	0.98	0.97	1.00
Zala	36	0.90	0.84	0.96	299	0.83	0.81	0.84

* SR: Standardized rate.

## Data Availability

The data presented in this study are available on request from the corresponding author. The data are not publicly available due to the request of the board members of the Hungarian National Ambulance Service.

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
