# Peer review of "Investigating the Geographic Disparities of Amenable Mortality and Related Ambulance Services in Hungary"

_ijerph, 2021, doi:10.3390/ijerph18031065_

Round 1
Reviewer 1 Report
The Authors have investigated very important problem of amenable mortality and related ambulance services in Hungary. The data generally come from National Ambulance Service and Central Statistical Office in Hungary – with all possible limitations related to the quality of data collection in large national medical services.
One remark: when our population globally is getting older analyses limited to the patients younger than 64 is at least debatable.
Second remark: I agree that acute myocardial infarction and stroke (both ischemic and haemorrhagic) are the most spectacular cause of death. Still other morbidities like pulmonary embolism are not rare. In all this case role of ambulance service is crucial in case of immediate diagnosis and direct patients for further medical treatment or to exclude life threatening condition.
Questions:
Considering the differences in rate of death, rate of deliveries among analysed counties are there in Hungary hospital networks dedicated for AMI and stroke patient treatment ?
In case of stroke have you investigated availability of angio TK diagnosis in all counties ?
Author Response
Dear Reviewer,
We would like to thank you for taking your time and for providing useful advice on improving the content of the manuscript. As required, we are uploading the ‘change tracked’ version of the revised manuscript.
Reviewer 1: The Authors have investigated very important problem of amenable mortality and related ambulance services in Hungary. The data generally come from National Ambulance Service and Central Statistical Office in Hungary – with all possible limitations related to the quality of data collection in large national medical services.
Reviewer 1.1: One remark: when our population globally is getting older analyses limited to the patients younger than 64 is at least debatable.
Response 1.1: We understand the concern you have brought up. In order to provide justification for the age criterion, we have written the following paragraph within the methods section (at line 105):
“The concept of avoidable mortality has undergone significant changes in content and terminology since its inception in 1976, including a change in the upper limit of the initial limit from 64 to 74 years [10, 20]. The use of a higher upper age limit was justified by the steady increase in life expectancy at birth and also by the fact that better results are being achieved at older age using innovative health interventions [10, 20]. Evidence shows however, that there are significant differences in the life expectancies of people in different European countries. Thus, for example, compared to the life expectancy of the population of the European Region, the Hungarian population lagged by almost 3 years in 2015 [21]. Moreover, in Hungary the gap between the life expectancy of men and women is also considerable. The difference was about 8.8 years in 2015 [22], but despite improving trends, it is still 6.47 years in 2019 (men: 72.86 years; women: 79.33 years) [23]. In addition, the outstandingly poor health outcomes of the Hungarian population, especially in the “early death stage of life (in the 25-64 age group)”, is also well known [21, 24, 25]. Consequently, due to the considerable lag in the life expectancy of the Hungarian population and its high early mortality, in the present analysis we decided to examine the deaths related to health services that occurred in the 15-64 age group.”
Reviewer 1.2: Second remark: I agree that acute myocardial infarction and stroke (both ischemic and haemorrhagic) are the most spectacular cause of death. Still other morbidities like pulmonary embolism are not rare. In all this case role of ambulance service is crucial in case of immediate diagnosis and direct patients for further medical treatment or to exclude life threatening condition.
Response 1.2: We agree with your statement. When discussing the research plan with the board members of the National Ambulance Service, they made the recommendation to investigate acute myocardial infarction and stroke. This is because these disease groups are relevant when assessing the performance of ambulance services.
Reviewer 1.3: Questions: Considering the differences in rate of death, rate of deliveries among analysed counties are there in Hungary hospital networks dedicated for AMI and stroke patient treatment?
Response 1.3: In Hungary all of the patients are delivered to one of the relevant centres. For the purpose of clarification, we added a paragraph about this question from line 75:
“The organization is completely government funded and all the ambulance deliveries are conducted by it. Hungary has 16 PCI (percutaneous coronary intervention) centers, 39 stroke centers, and out of all the centers 6 also has mechanical thrombectomy capability. In the past few decades the National Ambulance Service has been informing and educating the citizens as well as the medical providers in Hungary to call immediately an ambulance if an acute coronary syndrome or stroke is suspected in order to minimize the prehospital delay of the reperfusion treatment. By using regularly updated standard operational procedures (SOP) all over the country the National Ambulance Service tries to provide the same quality of service, resulting in equal recovery chances to all the patients.”
Reviewer 1.4: In case of stroke have you investigated availability of angio TK diagnosis in all counties?
Response 1.4: Every county in Hungary has at least one stroke center. All stroke centers have a wide range of diagnostical resources, by latest international guidelines for example small bore multi-slice CT scanners and Computed Tomography Angiography (CTA) to visualize the arteries and veins.
Also, we have asked one of our colleagues who speaks fluently English to carefully read the manuscript and check for grammatical errors. All the necessary changes were made with track changes.
We hope that these changes and responses are adequate for you.
Sincerely yours,
The authors of the manuscript

Reviewer 2 Report
Review report
Investigating the Geographic Disparities of Amenable Mortality and Related Ambulance Services in Hungary
This paper explores how preventable mortality may be related to ambulance services at a county level in Hungary.
Thank you for providing me the opportunity to read this manuscript. While I believe the issue of preventable mortality is indeed a major public health interest, I have identified a number of major issues that I believe the authors need to address before the paper can be reconsidered for publication.
First, the authors suggest that the rate of ambulance deliveries is a proxy for… what? Is it for the availability of services? Or maybe to the quality of services? This is not explained in the text. Does all counties is Hungary has the same access to Ambulance services? Is this service provided as part of public insurance or is it a private service? (and thus costly and affected by the economic ability of a person or household).
Furthermore, mortality from the conditions that were examined can be influenced by sociodemographic and economic characteristics of the patients (other than age and gender) – which to my best understanding were not taken into account in the current analysis. What about the health status of these patients? Being younger than 64 doesn’t guarantee good health.
I think all of the abovementioned questions/issues need to be thoroughly addressed.
Additional comments:
- Please provide a definition for ‘amendable mortality’, the definition I know is different…
- The authors state “Residents over the age of 64 were excluded in our study because we only focused on preventable mortality among adults”. This makes no sense to me – aren’t persons older than 64 adults? And surely you can agree that deaths among persons older than 64 can still be prevented.
- Table 1-4. Please provide the data as n(%).
- What are the average mortality rates you compare your data to? It’s not mentioned in the text.
- The discussion and conclusions sections are rather short and do not provide any in-depth discussion into the results.
Author Response
Dear Reviewer,
We would like to thank you for taking your time and for providing useful advice on improving the content of the manuscript. As required, we are uploading the ‘change tracked’ version of the revised manuscript.
Reviewer 2: Investigating the Geographic Disparities of Amenable Mortality and Related Ambulance Services in Hungary. This paper explores how preventable mortality may be related to ambulance services at a county level in Hungary. Thank you for providing me the opportunity to read this manuscript. While I believe the issue of preventable mortality is indeed a major public health interest, I have identified a number of major issues that I believe the authors need to address before the paper can be reconsidered for publication.
Reviewer 2.1: First, the authors suggest that the rate of ambulance deliveries is a proxy for… what? Is it for the availability of services? Or maybe to the quality of services? This is not explained in the text.
Response 2.1: We agree, that this should have been explained in the original manuscript. Before describing the goal of the study (at line 89) we now make this explanation as follows:
“Beyond these factors, however, the service provider must strive to reduce geographic inequality as much as possible. This can only be achieved, if the rate of ambulance deliveries – which is a reflection of the availability of services provided by the National Ambulance Service – and the health needs of the general population are in synch with each other.”
Reviewer 2.2: Does all counties is Hungary has the same access to Ambulance services? Is this service provided as part of public insurance or is it a private service? (and thus costly and affected by the economic ability of a person or household).
Response 2.2: Everyone who is in Hungary - regardless of citizenship or insurance status - is entitled to ambulance transportation. This means that income does not affect access to ambulance services. For clarification we added this complementary sentence in the manuscript at line 75: “The organization is completely government funded and all the ambulance deliveries are conducted by it.”
Reviewer 2.3: Furthermore, mortality from the conditions that were examined can be influenced by sociodemographic and economic characteristics of the patients (other than age and gender) – which to my best understanding were not taken into account in the current analysis. What about the health status of these patients?
Response 2.3: We agree that sociodemographic and economic characteristics are important factors when assessing both mortality and any form of healthcare service. While obtaining these data at a county level from the Hungarian Central Statistical Office would have been possible, the Hungarian National Ambulance Service does not include such data when delivering patients. Thus, for uniform methodology, we decided not to include these factors in the statistical analysis. This statement is also true for the general health status of the patients.
Reviewer 2.4: Being younger than 64 doesn’t guarantee good health. I think all of the abovementioned questions/issues need to be thoroughly addressed.
Response 2.4: We agree with your remark. In order to provide justification for the age criterion, we have written the following paragraph within the methods section (at line 105):
“The concept of avoidable mortality has undergone significant changes in content and terminology since its inception in 1976, including a change in the upper limit of the initial limit from 64 to 74 years [10, 20]. The use of a higher upper age limit was justified by the steady increase in life expectancy at birth and also by the fact that better results are being achieved at older age using innovative health interventions [10, 20]. Evidence shows however, that there are significant differences in the life expectancies of people in different European countries. Thus, for example, compared to the life expectancy of the population of the European Region, the Hungarian population lagged by almost 3 years in 2015 [21]. Moreover, in Hungary the gap between the life expectancy of men and women is also considerable. The difference was about 8.8 years in 2015 [22], but despite improving trends, it is still 6.47 years in 2019 (men: 72.86 years; women: 79.33 years) [23]. In addition, the outstandingly poor health outcomes of the Hungarian population, especially in the “early death stage of life (in the 25-64 age group)”, is also well known [21, 24, 25]. Consequently, due to the considerable lag in the life expectancy of the Hungarian population and its high early mortality, in the present analysis we decided to examine the deaths related to health services that occurred in the 15-64 age group.”
Also, we have asked one of our colleagues who speaks fluently English to carefully read the manuscript and check for grammatical errors. All the necessary changes were made with track changes.
We hope that these changes and responses are adequate for you.
Sincerely yours,
The authors of the manuscript

Reviewer 3 Report
Summary
The paper presents the findings of a study about the relationship between amenable mortality and ambulance transports from the point of the geographic disparities. The study aimed to characterize the differences in amenable mortality related to three main disease groups by comparing the death rates and the ambulance transports of various counties with the national average in Hungary. The study found significant differences between the counties, and by these results the authors would like to call the attention of health politicians/decision makers for the necessity of changes to provide equal access to the health services in all counties.
Comments and suggestions
The authors provided a brief overview about the topic in the Introduction. I think that the clear definition of amenable mortality (including the concept of avoidable mortality and its two parts such as amenable and preventable deaths) is missing from the introduction. The use of the appropriate expressions is also important (do not replace “amenable” with “preventable” – e.g., in line 57 – it may be better to use avoidable); the authors have to check it in the whole paper.
The detailed paragraph about the history of ambulance service is not essential, on the other hand some national data about the diseases under study can support the importance of the research.
The interpretation of the results is appropriate, the tables and the short explanations about the significant differences are clear.
Discussion: The study was done in Hungary; what is the situation in other European countries? What kind of ambulance service-related interventions can be suggested on the base of these data? Finally, the limitations of the study are not mentioned by the authors (e.g., the multifactorial aetiology of the diseases, the access to the ambulance services, urban-rural differences).
Author Response
Dear Reviewer,
We would like to thank you for taking your time and for providing useful advice on improving the content of the manuscript. As required, we are uploading the ‘change tracked’ version of the revised manuscript.
Reviewer 3: Summary: The paper presents the findings of a study about the relationship between amenable mortality and ambulance transports from the point of the geographic disparities. The study aimed to characterize the differences in amenable mortality related to three main disease groups by comparing the death rates and the ambulance transports of various counties with the national average in Hungary. The study found significant differences between the counties, and by these results the authors would like to call the attention of health politicians/decision makers for the necessity of changes to provide equal access to the health services in all counties.
Reviewer 3.1: Comments and suggestions: The authors provided a brief overview about the topic in the Introduction. I think that the clear definition of amenable mortality (including the concept of avoidable mortality and its two parts such as amenable and preventable deaths) is missing from the introduction.
Response 3.1: Thank you for your remark. We added the definitions of amenable, preventable and avoidable mortality in the first paragraph of the introduction (at line 46):
“A death can be considered as amenable if it could have been avoided through optimal quality health care” [3]. It should not be confused with preventable mortality, which is ”…broader and includes deaths which could have been avoided by public health interventions focusing on wider determinants of public health, such as behavior and lifestyle factors, socioeconomic status and environmental factors” [3]. The combination of amenable mortality and preventable mortality is called avoidable mortality.
Reviewer 3.2: The use of the appropriate expressions is also important (do not replace “amenable” with “preventable” – e.g., in line 57 – it may be better to use avoidable); the authors have to check it in the whole paper.
Response 3.2: Based on the definitions, “preventable” was used incorrectly in the manuscript. This was an error from our part. We have now replaced it “preventable” with “avoidable” at lines 57 and 104. At line 59 ”preventable deaths” was corrected to “amenable deaths”.
Reviewer 3.3: The detailed paragraph about the history of ambulance service is not essential, on the other hand some national data about the diseases under study can support the importance of the research.
Response 3.3: We have considerably shortened the section regarding the history of the ambulance service (see lines: 73-82). Also, we added some paragraphs in the introduction about disease data (see lines 57):
“Also, by the latest accessible Eurostat data, Hungary is among the most afflicted EU countries in cardiovascular diseases. After Lithuania, Hungary has some of the highest standardized death rates for ischemic heart diseases. Hungary also has the largest number of self-reported hypertensive diseases in the whole EU [11].”
Reviewer 3.4: The interpretation of the results is appropriate, the tables and the short explanations about the significant differences are clear.
Response 3.4: We appreciate this positive remark.
Reviewer 3.5: Discussion: The study was done in Hungary; what is the situation in other European countries?
Response 3.5: We are unable to compare the results with other European countries, because while in Hungary there is a single organization responsible for all ambulance deliveries, other European countries organize their ambulance service at a regional level. Also, no similar study has yet been conducted in Europe.
Reviewer 3.6: What kind of ambulance service-related interventions can be suggested on the base of these data?
Response 3.6: In the conclusion we now describe in detail the interventions that can be made based on the results as follows:
“The principle of equal access is an important health policy objective [28]. Thus, Hungarian decision-makers should create policies and an overall strategy based on the obtained results. For example, better allocation of resources between the counties, the expansion of priority units in some counties, or creating efficient health screenings in the underperforming areas.
In order to reduce inequalities, an important action would be to train and educate the general population regarding the symptoms of various diseases, the encouragement of making rescue calls and providing first aid. The feedbacks of these actions should be shared with both the ambulance service's rescue and management staff and, if necessary, professional training should be provided to achieve better results. Based on the results of this study, the National Ambulance Service plans to introduce a new, uniform and professional rescue manager interview protocol, as well as to issue a new organization form regarding the vehicle fleet, location and necessary modification of ambulance vehicles.
Such implemented policies should be examined, and the impact of the interventions should be evaluated as well. Lessons learned from successful interventions can and should be shared with rescue services in other countries. Furthermore, we encourage other researchers to conduct similar studies, as it is probable, that similar differences can be identified.”
Reviewer 3.7: Finally, the limitations of the study are not mentioned by the authors (e.g., the multifactorial aetiology of the diseases, the access to the ambulance services, urban-rural differences).
Response 3.7: Thank you for your comment, as this is a very important shortcoming. Thus, the following paragraph was included at the very end of the discussion:
“Within the limits of our study, it should be mentioned that the multifactorial etiology of the cardiovascular diseases studied was not considered in the study. The study was conducted at the county level as the data of the Hungarian ambulance service were available in this form, so the study could not analyze the municipalities according to the degree of urbanization and their geographical location. An additional limitation is that although the Hungarian ambulance service provides uniform coverage of patient transportation in the country, the related healthcare providers in the geographical areas may not be of exactly the same professional standard and equipment throughout the country, which may affect avoidable mortality rates. Thus, the aim of our study was to present a result that draws attention to the importance of the efficient operation of the ambulance service and the existence of differences that can be reduced.”
Also, we have asked one of our colleagues who speaks fluently English to carefully read the manuscript and check for grammatical errors. All the necessary changes were made with track changes.
We hope that these changes and responses are adequate for you.
Sincerely yours,
The authors of the manuscript

Round 2
Reviewer 2 Report
Thank you for the opportunity to review this revised version of the manuscript. I still feel the discussion section (and also the part that discusses the limitation of the current analysis) should be expanded to deliver a clear message to the redears.
Table 1 is still in absolute numbers which I find to be less informative.
Author Response
Dear Reviewer,
Thank you for your time and comments. The necessary changes were made using ‘Track Changes’. In order to easily follow the changes currently made, we have accepted all the changes previously made in the first revision, thus, only the latest alterations can be seen in the second revision of the manuscript.
Reviewer 2.1: Thank you for the opportunity to review this revised version of the manuscript. I still feel the discussion section (and also the part that discusses the limitation of the current analysis) should be expanded to deliver a clear message to the readers.
Response 2.1: We would like to thank you for reviewing the revised version of the manuscript. At the discussion we now describe how the Hungarian National Ambulance Service could benefit from the findings of the study (see lines: 204-211):
“This study also draws the attention of the management of the National Ambulance Service to further emphasize the utilization of check-list type disease debriefing and the supervision of this procedure during the training of rescue managers and those taking part in rescue operations in order to be able to make appropriate decisions on the place of dispatch and the level of competency needed from the dispatched rescue unit, thus, ensuring the highest quality of care possible.
Furthermore, based on the results of the analysis the operative team of the National Ambulance Service can formulate a proposal on the appropriate allocation of ambulance units to the top management, in order to aid the implementation of a rational and professional vehicle reallocation.”
Also, when describing the limitation, we now explain what ‘multifactorial etiology of the cardiovascular diseases’ means in our study (see lines: 213-215):
“Our mortality and ambulance delivery results at the county level not necessarily mean the same outcomes in individuals. The detected results by our investigation do not prove a causal relationship between mortality and ambulance deliveries.”
Reviewer 2.2: Table 1 is still in absolute numbers which I find to be less informative.
Response 2.2: Due to the size of the table, we could not include both the absolute numbers (N) and the percentages (%). As a compromise, the national total is shown at the top of the table as absolute numbers, while the values of the counties are shown as percentages (see Table 1).
Finally, some minor typos were also corrected using ‘Track Changes’.
We hope that these changes and responses are adequate for you.
Sincerely yours,
The Authors
